# INFORMATION STRUCTURE IN LARGE LANGUAGE MODELS

## ABSTRACT

Despite the widespread use of large language models, we still lack unified notation for thinking about and describing their representational spaces. This limits our ability to understand how they work. Ideally we would understand how their representations are structured, how that structure emerges over training, and what kinds of structures are desirable. Unfortunately we as humans tend not to have strong intuitions about high-dimensional vector spaces. Here we propose an information theoretic approach to quantifying structure in deep-learning models. We introduce a novel method for estimating the entropy of vector spaces, and use it to quantify the amount of information in the model we can explain with a set of labels. This can show when regularities emerge in representation space with respect to token, bigram, and trigram information in the input. As these models are learning from human language data, we formalise this in terms of 3 linguistically derived quantities: regularity, variation, and disentanglement. These show how larger models become proportionally more disentangled. We also are able to predict downstream task performance on GLUE benchmarks based on representational structure at the end of pre-training but before fine tuning.

## 1 INTRODUCTION

Despite the remarkable performance of large language models (Brown, 2020; Dubey et al., 2024), and their widespread use we still lack unified notation for thinking about and describing their representational spaces. We lack methods to reliably describe how their representations are structured, how that structure emerges over training, and what kinds of structures are desirable. This should be of concern to us for practical reasons - it makes it difficult to make design decisions when we don't have a clear picture of how they effect representational space - but also for broader for social reasons. Most people in the US and UK come into contact with an NLP system multiple times a day without realising (Kennedy et al., 2023). Given their increasing ubiquity our limited ability to account for the information they have learned and how that information is structured is worrying.

Part of the reason for this is that their representations are continuous, and we as humans tend not to have strong intuitions about high-dimensional vector spaces. Existing work in interpreting large language models describes phases of training in terms of model behaviour (e.g. Marvin, 2018; Blevins et al., 2018; Dziri et al., 2024) like analysing when they begin to generalise robustly - or *grok* (Power et al., 2022; Merrill et al., 2023). Alternately work uses parametric methods like probing, leveraging a separate model to describe the first (Hupkes et al., 2018; Voita & Titov, 2020; Pimentel et al., 2020). We focus instead on giving a representational account of what training looks like, using information theoretic measures of representational space to quantify how structured representation spaces are in large language models, and what kinds of structure matter for generalisation. Ideally we need a way of thinking about deep-learning models in the general case that allows us to: 1) Describe structure in representation space, and what structures drive generalisation, 2) Clearly relate these to relevant work in linguistics and the cognitive sciences, 3) Quantify structure with methods that are efficient enough to apply the same analyses to models of any size, throughout training, and 4) Meaningfully compare models of different sizes, trained with different objectives.

In an effort to do this, we look at deep-learning models as member of a more general class: mappings. Models map between their inputs and representational space, and are comprised of a sequence of linear and non-linear mappings. Here we quantify structure in the mappings learned by large lan-

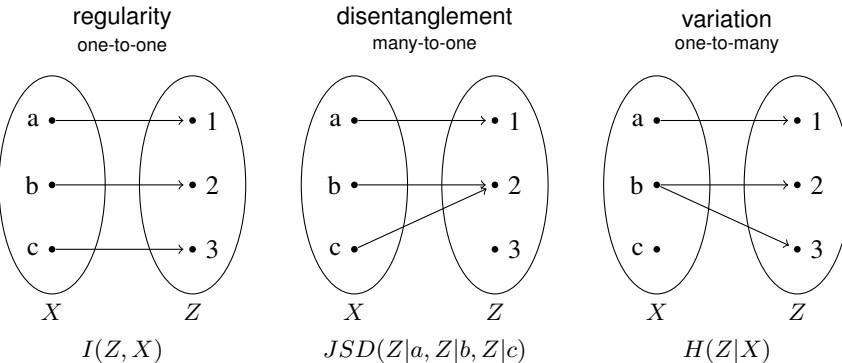

Figure 1: Three basic kinds of mapping structure we consider here, labelled with their linguistic analog, and the information theoretic quantity we introduce to measure them in section 4. Note that we show part of the mapping ($a \to 1$) as regular in all cases because the mappings we consider exhibit a combination of all 3 structures. As such we assess the *degree* of each structure, not whether or not it exists. Variation (one-to-many) is possible here because our $X$ contains instances of the same token in different sentences, meaning $b \to 2$ and $b \to 3$ reflects b in different contexts which are not shown here for brevity.

guage models while drawing parallels to a reference mapping about which we have strong intuitions for what structure looks like - unlike high-dimensional vector spaces - and which is related to the domain in which our models are trained: natural language.

At its core language is a mapping - relating real-world objects, concepts, and events, to words, constructions, and phrases which refer to them (de Saussure, 1916). While many natural communication systems fit this bill, language is unique amongst them (Hockett, 1960). It's learned from a finite sample, generalises readily to novel concepts and contexts, with *system level* structures that provide us a system simple enough to be learned by children, but expressive enough to describe the universe. This parallels our desiderata for mappings in deep-learning models which need to be learned from finite data, able to generalise, and expressive enough to describe the world from which their training data is drawn. We look at whether *system level* structures emerge in representation spaces learned by large language models; first introducing basic kinds of structure in a mapping, relating them to their analogs in linguistics, before quantifying each of them information-theoretically.

We build on the framework for interpretability introduced in Conklin & Smith (2024), redefining some of their measures, and extending it to large language models. To do this we also introduce a novel method for highly-performant entropy estimation in vector space - ***soft entropy***. This approach is similar to discretisation based methods used to analyse deep-learning (Shwartz-Ziv & Tishby, 2017b; Goldfeld et al., 2018), but fully differentiable, less affected by hyper-parameter settings, and dramatically more memory & compute efficient. Additionally the estimator can easily be applied at different levels of abstraction like model, layer, and subspace - this broken-down estimate enables direct comparison between different model sizes. We use soft entropy to quantify structure in language models ranging from 14 million to 12 billion parameters, looking at when system-level structure emerges during training, how scaling affects representation structure, and what kinds of structure drive generalisation. Our analysis is able to predict downstream performance on GLUE benchmarks based only on a models' representations at the end of pre-training (before 2 million steps of fine-tuning). To summarise our core contributions, this paper:

- Frames structure in large language models in terms of related notions of structure from linguistics and information theory

- Introduces a novel method for entropy estimation of continuous spaces, that's fast, efficient and differentiable

- Shows how scaling a model's hidden dimension, or number of layers affect representational structure

- Correlates representation structure at the end of pre-training with performance downstream after fine-tuning

## 2 RELATED WORK

Our work is related to a long history of research in NLP trying to work identify correspondences between linguistic structures in training data and representations or behaviours (Shi et al., 2016; Belinkov et al., 2017; Marvin, 2018; Blevins et al., 2018; Dziri et al., 2024). It's particularly closely related to probing (Hupkes et al., 2018; Pimentel et al., 2020) which trains a classifier to predict labels from a larger model's representations. MDL probing (Voita & Titov, 2020) also includes a notion of regularity in terms of the complexity of the probe required to recover the labels. Given we quantify structure in the mapping between labels and representations directly our work represents a non-parametric approach to probing. The analysis here is also related to work in language emergence which looks at the languages that emerge between models in a multi-agent setting. A variety of quantifications of linguistic structure have been proposed for that domain that leverage similar intuitions to the ones used here (Brighton et al., 2005; Lazaridou et al., 2018; Resnick et al., 2020; Chaabouni et al., 2020; Conklin & Smith, 2022)

There's also existing work that tries to characterise training dynamics information theoretically (Tishby & Zaslavsky, 2015; Shwartz-Ziv & Tishby, 2017a; Goldfeld et al., 2018; Saxe et al., 2019), however these are largely theoretical works and/or applied to feed-forward networks on tasks like digit classification. (Conklin & Smith, 2024) applies information theoretic methods to transformers trained on a single task - but uses dimension-wise discretisation which is difficult to scale. Our approach to estimating entropy is similar to the limiting density of discrete points (Jaynes, 1957) and is related to kernel density estimation (Parzen, 1962) in the way it relates discrete points to a continuous function.

## 3 IDENTIFYING STRUCTURE IN MAPPINGS

We consider 3 basic structures in a mapping between two spaces: *one-to-one*, *many-to-one*, and *one-to-many*. These are related to linguistic concepts of regularity, disentanglement, and variation respectively. In a model we quantify these properties between labels for a model's input and the corresponding representations. The labels can be any that you have for an input sentence, in the experiments here we use ones that you get for free with any text data: token, bigram, and trigram. This lets us look at lexical and contextual information in the model and shows the generality of the approach. If you had data labeled with parts of speech you could look at how syntactic information is represented, or a set of sentences labelled as True or False would let you look at factuality – the point being given a set of labels for the input we quantify structure in representation space with respect to them.

$$Z = [z_a^k : \forall z_a^k \in f(x^k) : \forall x^k \in X] \quad (1a) \qquad Z|\text{label} = [z_a^k \text{ if } a = \text{label} : \forall z_a^k \in Z] \quad (1b)$$

To formalise this in terms of the transformer language models we're working with - we look at the mapping at the token level. We have a model $f$ that maps a set of sentences $X$ to representational space $Z$. For each sentence $x^k \in X$, the model takes as input a sequence of tokens $t_a^k, t_b^k, t_c^k... \in x^k$ and returns a sequence of vectors $z_a^k, z_b^k, z_c^k... \in Z^k$ where $z_a^k$ is the vector corresponding to token $a$ when it occurs in sentence $k$. While each sequence $Z^k$ is of variable length, the individual vectors are the same size. We can create a list $Z$ of all token representations from all sentences in the dataset, or a list of all tokens corresponding to a given label $Z|label$. This means when we look at bigram or trigram information, we're labelling the representation $z_a^k$ with $(a, b)$ or $(a, b, c)$. The next section explains how we estimate entropy in vector space, first we walk through the kinds of structure we measure. The estimation procedure gives us a categorical distribution which describes vector space $P(Z)$ used below.

$$\text{variation}(Z, \text{set}) = \frac{1}{|\text{set}|} \sum_{\text{label}}^{\text{set}} H(Z|\text{label}) \tag{2}$$

**Variation**   describes how much representations for a label vary in representation space. In the token case this reflects whether a model learns a single context independent representation of the token or a different representation for every sentence it occurs in. We can quantify this in terms of the conditional entropy of space given a label. The resulting quantity is related to intrinsic dimensionality, reflecting how much of representational space is used to represent a given feature in the input, but faster to compute given it requires no pairwise comparisons. We bound this and the regularity measure to lie between 0 and 1 to aid interpretation.

$$\text{regularity}(Z, \text{set}) = \frac{1}{|\text{set}|} \sum_{\text{label}}^{\text{set}} H(Z) - H(Z|\text{label}) \tag{3}$$

**Regularity**   reflects the amount of variation in representation space we can explain by knowing a label. Based on mutual information it's the difference between overall variation in the space H(Z) and the variance in representations for a given label H(Z—label). It reflects how monotonically aligned representation space is with that label. In language regularity is often measured similarly (Smith & Wonnacott, 2010; Ferdinand et al., 2019; Carr et al., 2020) and is used to quantify how syntactically structured a system is. We bound this measure by the entropy of a uniform distribution

$$\text{disentanglement}(Z, \text{set}) = H(M) - \sum_{\text{label}}^{\text{set}} P(\text{label})H(Z|\text{label}) \tag{4}$$

**Disentanglement**   measures whether clusters corresponding to labels within a set are separable, like if different tokens or bigrams are represented in different parts of space. We estimate this with a multi-variate Jensen-Shannon divergence. This requires a mixture distribution $M$ computed by taking the mean of individual label distributions weighted by the probability of the label $M \propto \sum_{\text{label}}^{\text{set}} P(\text{label})P(Z|\text{label})$. The result is bounded by the entropy of the mixture distribution, which we use to normalise the measure so that as values approach 1 labels are maximally separable in space, and as it approaches 0 all labels in a set occupy the same region of space. This is related to previous measures of entanglement (Chen et al., 2018; Conklin & Smith, 2024) but is faster to compute, and allows labels to contribute proportionally to the estimate based on their probability.

## 4   SOFT ENTROPY ESTIMATION

There are few approaches to entropy estimation that are sufficiently fast and memory efficient to be applied to large language models. This is frustrating given information theoretic tools are well suited to quantifying complex structures in distributed systems. With soft entropy we introduce an approach that prioritises efficiency, while performing comparably to existing methods. It's worth noting that we focus on estimating the *discrete* entropy rather than differential entropy. We draw inspiration from Jaynes (1957), who notes differential entropy is not the true continuous analog of discrete entropy and proposes the limiting density of discrete points as an alternative. This takes entropy to be the divergence between a distribution and an invariant measure (usually a uniform distribution over the same support); it reflects how 'non-uniform' a distribution is. Our method follows this intuition sampling points uniformly across space, and comparing them with samples from the model.

We define a mapping between real-valued space and information space, creating a categorical distribution that describes a model's representation space. Our estimator returns the entropy of the descriptor distribution a quantity we call *soft entropy* - distinct from the differential entropy of the space. This process is akin to 'plug-in' estimation, where you first fit a distribution then estimate it's entropy - except here the distribution we 'fit' is categorical. Existing approaches to estimating entropy of vector space often rely on discretisation with clustering, or binning, the approach described

here can be seen as a differentiable relaxation of these methods. We do benchmark our approach on reference distributions with known entropy, and show that it performs similarly to clustering and discretisation, these results can be found in appendix A.1.

## 4.1 FORMALISATION

Given a set of representations $Z$ with dimensions batch size $bs$ by hidden size $h$ we take the euclidean norm, so they lie on the unit sphere. We then sample points uniformly from the surface of the unit sphere, by drawing $n$ samples from a standard normal and taking their euclidean norm. The resulting points $S$ have dimensions $h \times n$ where $n$ is a hyperparameter controlling the number of points. To assess how close each representation is to each point we take the dot product between $Z$ and $S$. The result is a cosine similarity, which we pass through a softmax to get a distribution over points for each representation with dimensions $bs \times n$. By summing over the batch dimension and re-normalising we get a single categorical distribution that describes the space $P(Z)$ with dimensionality $1 \times n$. To get a binning based estimate we could treat each point as the center of a bin, and assign representations to the point they're closest to, rather than normalising distances.

$$P(Z) \propto \sum \operatorname*{softmax}\left( \underset{bs \times h}{\frac{Z}{|Z|}} \cdot \underset{h \times n}{\frac{S}{|S|}} \right) \qquad (5)$$
$$\underset{1 \times n}{}$$

Because this gives us a categorical distribution, estimation of information-theoretic quantities is straightforward. Soft Entropy of the space follows the equation for shannon entropy: $H(Z) = -P(Z) \log P(Z)$. We can also quantify entropy in subspaces, as opposed to the entire space by applying the estimator in a multi-headed arrangement. We reshape the representations from $bs \times hidden$ to $bs \times head \times \frac{hidden}{heads}$ and the points to $\frac{hidden}{heads} \times heads \times bins$. This allows us to estimate entropy per-head and mean across them.

H(Z) reflects how uniformly distributed representations are across angles with respect to the origin. It is maximised when representations are uniformly distributed across all 360 degrees, and approaches 0 as representations cluster across an increasingly small subset of angles. This quantity is related to anisiotropy, where representations lie in a narrow cone relative to the origin, but is dramatically faster to compute than taking pairwise cosine similarities between all representations. We draw a parallel between this measure and clustering based estimates of entropy, where representations are first clustered, then discretised. Here when we project points to the unit sphere we make representations with high cosine similarity, close to each other. To get a clustering estimate we could replace the events in the categorical distribution $P(Z)$ with clusters on the unit sphere rather than uniformly sampled points. In practice sampling points is substantively faster than performing clustering.

## 4.2 PARAMETERS & COMPUTATIONAL EFFICIENCY

Like how discretisation methods are sensitive to the number of bins used, soft entropy is sensitive to number of 'points' although less so than the discrete case: if two representations are close to each other they can't be split into separate 'bins,' given we get a distribution over points for each representation rather than assigning it to one. This means increasing number of points doesn't necessarily have a detrimental effect on mutual information and divergences but can still inflate the estimate. In the experiments presented here we use 50 bins unless otherwise noted. Additionally a softmax is not invariant to linear transformations and the distances from the dot product are bounded between -1 and 1, this can mean the default estimate is relatively high. After testing on reference distributions we opt to rescale the distances to lie between -100 and 100. This scaling factor is a parameter, like the bandwith parameter in kernel density estimation (Parzen, 1962), controlling the spread with respect to each point.

Our methods map representational space to a categorical distribution using a single dot product, softmax, and summation. These operations are differentiable, memory efficient, fast, and parallelisable. This process is non-parametric, requires no clustering, and is substantially more memory efficient than binning based approaches to entropy estimation which usually requires a step where representations are $bs \times seq \times hidden \times bins$ - using 100 bins on a model with 4096 dimensional spaces proves problematic.

## 5 EXPERIMENTS

We use our measures of structure in a mapping, and soft entropy estimation to analyse properties of large language models in three ways. First we look at the how structure develops over the course of training in an encoder-only transformer, analysing 5 different initialisation of BERT over 2 million training steps in section 5.2. In section 5.3 we look at how model size affects representational space in both encoder and decoder only models. Comparing structures inside decoder-only models ranging from 14 million parameters to 12 billion from the Pythia collection of models (Biderman et al., 2023). We also look at different sizes of BERT released in (Turc et al., 2019), which allows us to make more precise comparisons varying number of layers, or hidden size independently, rather than just overall parameter count. Finally in section 5.4 we look at the relationship between representation structure and downstream task performance. We use the Multiberts (Sellam et al., 2021), 25 BERT base models that differ only in their initialisation correlating their representation structure at the end of pretraining with performance 2 million steps of fine-tuning later.

### 5.1 ESTIMATING ENTROPY TO ENABLE MODEL COMPARISONS

Making fair comparisons between different models is often challenging given differences in number of layers and dimensionalities. Previous information theoretic analyses in deep-learning often report estimates for each layer separately (e.g. Shwartz-Ziv & Tishby, 2017b; Voita et al., 2021), which can make overall interpretation and comparison difficult. Instead we look at a model's hidden state as a single random variable distributed across layers. In practice though larger hidden states will have more information, what we want in order to make fair comparisons is a relative entropy estimate, reflecting how much information a representation space encodes proportional to its size.

To this end we report two different quantities, *layer entropy* and *subspace entropy*. For layer entropy we compute an estimate at each layer, then aggregate mean across them. This lets us directly compare models of the same dimensionalities but differing depths. For subspace entropy we apply the soft entropy estimator in a multi-headed arrangement as described in section 4.1. This lets us break representation spaces into lower dimensional subspaces, in the results here subspace entropy is computed over 32-dimensional spaces across every layer in the model then aggregated. This lets us compare entropies over the same sizes subspace for models with different overall dimensionalities. While breaking a vector into subspaces may break some cross-dimensional dependencies we believe that this effect is relatively small - results testing this on sample distributions are included with other entropy estimate benchmarking in the appendix A.2.

### 5.2 WHEN STRUCTURE EMERGES DURING TRAINING

We look at 5 different initialisations of BERT over the course of 2 million training steps (model checkpoints also released as part of Sellam et al. (2021)). At each checkpoint we compute our 3 structure measures with respect to token, bigram and trigram labels from the wikipedia data. We choose to use these labels because they are known for virtually every text dataset that's fed into a model.

Main findings are shown in figure 2. Overall trajectories for each measure are remarkably similar to the phases of training described in Conklin & Smith (2024), which applied a similar analysis to 3 layer encode-decoder transformers trained on a single semantic-parsing task - suggesting some generality to this characterisation of training dynamics in deep-learning. At the start of training ($< 100,000$ steps) representations quickly align with token-level information, with distinct tokens becoming represented in distinct, disentangled parts of space. Past this point the dynamic shifts as representations begin to contextualise. Token disentanglement drops significantly, while bigram and trigram disentanglement increase. These likely contrast because in order to better represent lower-level information like bigrams, separate tokens need to spread out (variation increases) and overlap (disentanglement decreases). This process of contextualisation is the defining dynamic of the majority of training. Unlike findings in Conklin & Smith (2024), later stages of training are not characterised by overall compression of the space (overall entropy decreasing), this may be a difference between single task models and LLMs or may reflect that BERT was substantively undertrained, as noted in Liu (2019).

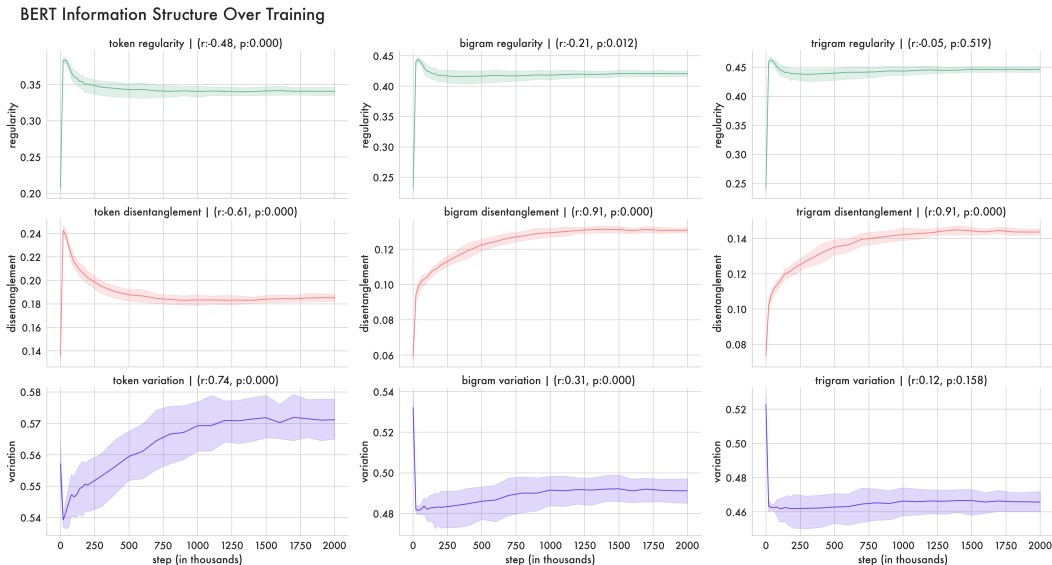

Figure 2: Information Structure with respect to 256,000 sentences from wikipedia over 2 million steps of training. Each line represents the mean of 5 different initialisations of BERT with shading representing 95% confidence intervals. Also included above each facet is a spearman correlation between x and y. Estimates here use layer entropy, given there's no need to compare different dimensionalities

## 5.3 MODEL SIZE CONDITIONS REPRESENTATIONAL STRUCTURE

How does scale affect representational structure? We look at this in both decoder-only and encoder only models again performing structure estimates using 256,000 sentences from english wikipedia, and labels for token, bigram, and trigram information. Figure 3 shows results for the decoder-only models, with both layer and subspace entropy reported. Both are reported for reference, and to give an intuition to how they relate - but as discussed above layer entropy does not allow a like-for-like comparison between different dimensionalities. As you would expect larger models have higher layer entropy - each layer of the 12b model has 5120 dimensions compared with 128 in the smallest, it would be surprising if they contained the same amount of information. Subspace entropy - which provides a more directly comparable estimate between model sizes - reveals a different pattern with the largest models beginning to compress their representations more, with the 12b version almost matching the subspace entropy of the smallest model. Because the representation space is larger information can be more distributed across space, meaning each subspace can compress more on a relative basis. We draw an analogy to Shannon's source coding model (Shannon, 1948) where meanings are mapped to signals; signal space has two key parameters - signal length and alphabet size. A smaller alphabet has less uncertainty, think of morse code with a binary alphabet where operators only need to tell the difference between a dot and dash. However smaller alphabets require a longer signal - sentences in morse code are far longer than in english - this is the tradeoff for more robust representations. In our subspace entropy analysis larger models have more subspaces, analogous to a longer signal. This can enable compression of each subspace like shrinking the alphabet at each character in a signal, which may help explain their improved performance.

The top plot in figure 3 plots the proportion of representation space that encodes token, bigram and trigram information and the information we can't explain in terms of any of the labels - the residual. This is estimated by comparing the regularity for each set of labels with the information left over which isn't regular with respect to any label. Looking at the subspace analysis larger models devote more of their representational space to contextual information, and less to token information. They also have more information we can't explain in terms of these labels. That could reflect other information from the training data not explainable in terms of lexical/contextual labels, or it could reflect artefacts not explainable in terms of any label. The middle plot shows disentanglement across model sizes, with larger models subspaces disentangling contextual information more.

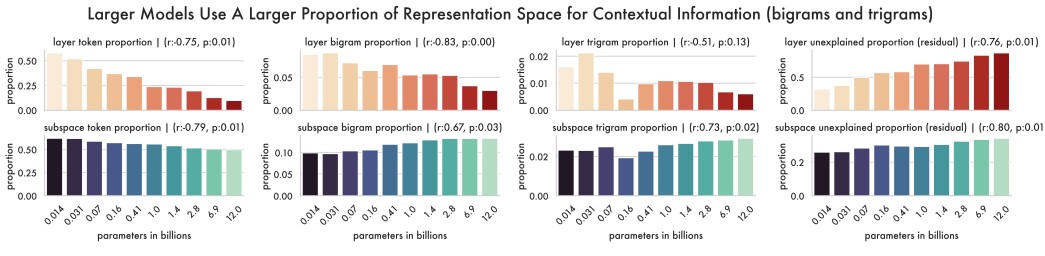

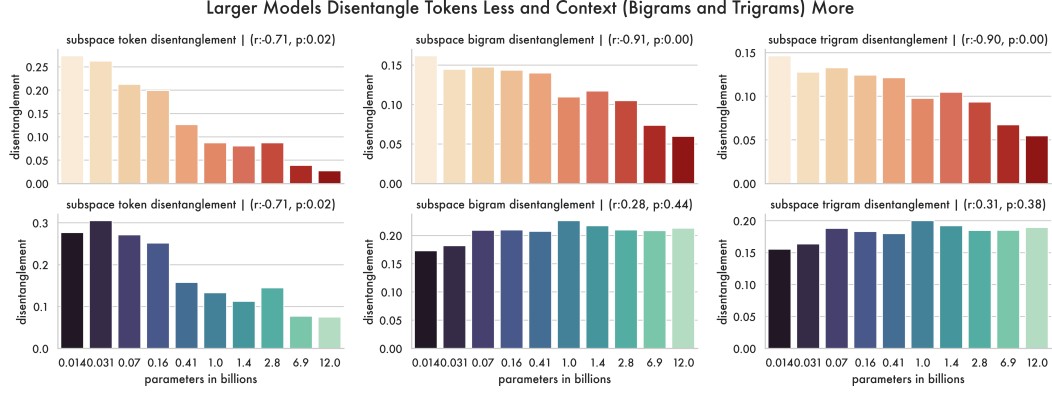

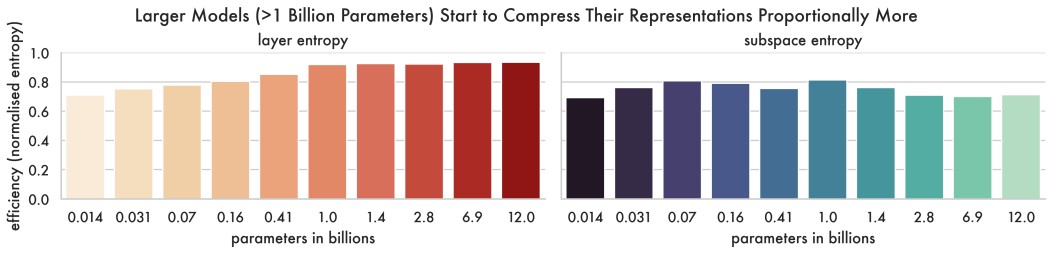

Figure 3: Analyses computed on pythia decoder-only models ranging from 14m parameters to 12 billion. Red/Orange bars show mean layer entropy, blue/green bars show mean subspace entropy. Above each plot is a spearman correlation between x and y **Top:** y-axis shows the proportions of representation space that encode token/bigram/trigram information for each model size(on the x-axis). Subspace entropy shows larger models use proportionally more space for token and bigram information. **Middle:** y-axis shows disentanglement for different model sizes. Subspace entropy shows bigrams are more disentangled in larger models **Bottom:** y-axis shows overall entropy of each model size. While layer entropy increases monotonically with size as expected - subspace entropy begins to compress in larger models.

An issue with the pythia suite of models is that while they differ in size, that difference is driven by changes in both depth and dimensionality[1]. In an effort to isolate the effects of these different kinds of scaling we use sets of BERT models released by Turc et al. (2019). Figure 4 shows effects on representational structure for models with a dimensionality of 768, but layers ranging from 2 to 12, and models with 12 layers but dimensionalities from 128 to 768. Overall both kinds of scaling have a similar effect, with dimensionality being much stronger than depth.

---

[1]It's also worth noting models also differ in the dimensionality of attention heads. this may have an interesting affect on structure but we lack controlled comparisons to draw conclusions.

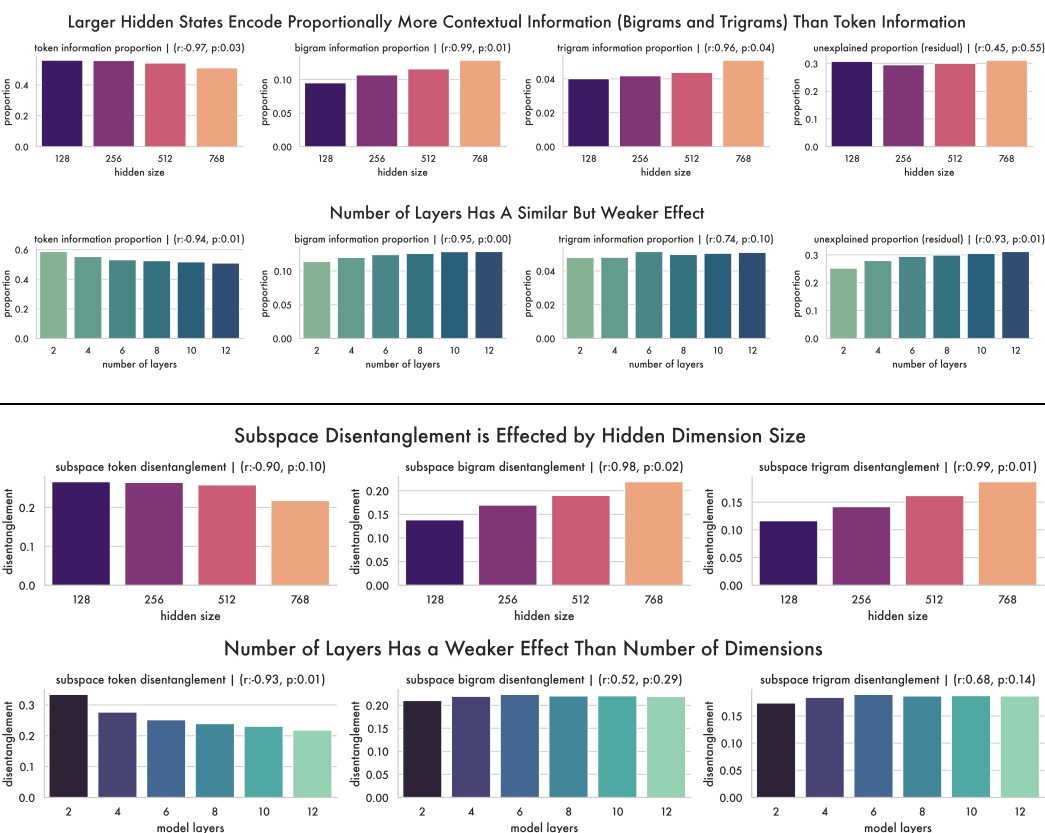

Figure 4: Scaling comparison of depth vs. dimensionalities on BERT models. All plots use subspace entropy, color reflects depth or dimension (both shown on the x axis) – atop plots is a spearman correlation between x and y **Top:** y-axis proportion of representation space that encodes token/bigram/trigram information **Bottom:** disentanglement of label information (y axis) across different sizes

## 5.4 PREDICTING DOWNSTREAM PERFORMANCE

We look at spearman correlation coefficients between structural properties of representation space and downstream task performance. In order to isolate as many variables as possible we use the Multibert models (Sellam et al., 2021) which is 25 different initialisations of BERT (Devlin, 2018). By comparing performance between models that differ only in terms of the random seed used to initialise them we can have some confidence that effects we measure between representational structure and downstream performance are likely driven by structure rather than model size, training data, or training objective. The Multiberts provide checkpoints at the end of pre-training, and evaluations for fine-tuned versions of each of these across the GLUE benchmarks (Wang, 2018). We take 2.5 million sentences sampled randomly from english wikipedia and compute our structure measures with respect to token, bigram, and trigram labels. We correlate representational structure at the end of pre-training with performance on GLUE tasks after fine-tuning. It is important to note that this means we are able to predict which of the models will do better on a downstream task before the models are fine-tuned for 2 million steps on data from that task. As far as we're aware this is the first analysis able to predict downstream performance from pre-training. Additionally the structure measures we use in this correlation are not estimated using data from those benchmarks. Despite the estimate using non-task data, on models 2 million steps of fine-tuning removed from evaluation we still find a number of significant correlations. That we can do this suggests our measures are contentful, and that representational structure has an effect on generalisation performance.

This analysis tells us two things, whether or not the data used to estimate structure is relevant to the downstream task - and if it is, how that information should be structured. Table 1 shows summary

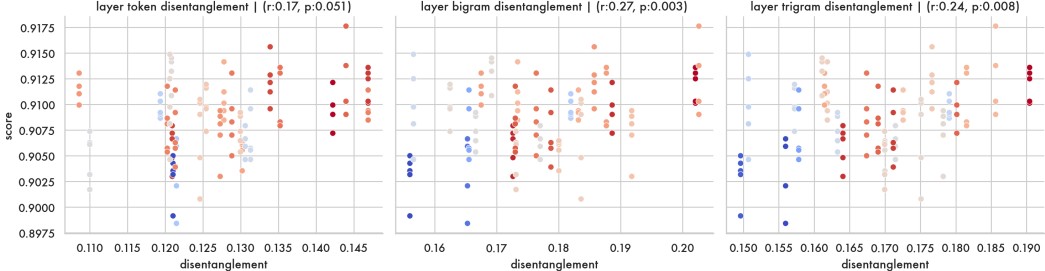

Figure 5: Scatterplots showing model performance on the QNLI benchmark (y-axis) vs. disentanglement at the token bigram and trigram level. As noted in table 1, on QNLI disentanglement with respect to these labels before fine-tuning correlates positively with performance after fine-tuning. Hue indicates variation, red indicating more variation, blue indicating less. As shown by the shift from blue to red as you look left to right, greater disentanglement allows for greater variation. Spearman correlation coefficients between x and y axes included above.

| sum across token/ bigram/trigram | STS-B | | MRPC | | MNLI | | CoLA | | QNLI | | QQP | | SST-2 | | RTE | |
|---|---|---|---|---|---|---|---|---|---|---|---|---|---|---|---|---|
| | r | p | r | p | r | p | r | p | r | p | r | p | r | p | r | p |
| regularity | 0.159 | 0.077 | n/s | n/s | -0.164 | 0.067 | n/s | n/s | 0.29 | 0.001 | 0.264 | 0.003 | -0.204 | 0.022 | n/s | n/s |
| disentanglement | 0.17 | 0.059 | n/s | n/s | n/s | n/s | n/s | n/s | 0.256 | 0.004 | 0.169 | 0.06 | -0.309 | 0.0 | n/s | n/s |
| variation | 0.201 | 0.024 | n/s | n/s | -0.287 | 0.001 | n/s | n/s | 0.307 | 0.0 | 0.233 | 0.009 | n/s | n/s | n/s | n/s |

Table 1: Summary Spearman Correlations between representational structure across 25 different initialisations of BERT at the end of pre-training (before fine-tuning) and downstream task performance on GLUE benchmarks (after 2M steps of fine-tuning). For readability correlations with p values > 0.1 are labelled as not significant (n/s). Positive correlations indicate models which are more structured with respect to the wikipedia data used to estimate the measures perform better. Negative correlations indicate models that preserve less information from wikipedia perform better. n/s indicates structure with respect to the data used is not relevant to this task's performance

correlations between pretraining structure and post-finetuning performance - for brevity values are summed across token/bigram/trigram levels before correlating, full correlations at all levels of analysis can be found in appendix A.3. Information structure with respect to wikipedia data - which is what our analysis provides - correlates positively with performance on STS-B, QNLI, QQP, for other tasks it is either slightly negative or not significant. Non significance suggests the data used to generate the estimate is unlikely to be relevant to the task. Figure 5 visualises the relationship between disentanglement and performance on QNLI, with higher disentanglement at all levels being related to performance, but with bigram and trigram being related more strongly.

## 6 CONCLUSION

We've introduced a set of measures for thinking about and describing structure in large language models information theoretically. This approach can show how representations become structured over the course of training, how that structure is influenced by model scale, and what structural properties correlate with downstream performance. It's backboned by a new scalable, performant, and differentiable approach to entropy estimation, that can be applied at the subspace level to enable like-to-like comparisons between models of different sizes. We related the structural properties found here to structures in linguistics, and Shannon's model of communication in an effort to contextualise these structures in terms of other areas of science. We think that continued work mapping large language models to spaces and measures about which we have stronger intuitions than vector space is crucial in helping us understand, interpret, and improve models going forward.

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

# A APPENDIX

## A.1 BENCHMARKING: SOFT ENTROPY VS. KMEANS CLUSTERING VS. DISCRETISATION

We compare our estimator (the two columns in the middle with soft in the title) with fully discretising the space and with kmeans clustering. Samples are drawn from a normal distribution with random covariance matrix. To compare the discretised estimator with the closed form differential entropy we use the histogram estimator of differential entropy to convert between them. This requires dividing the closed form by dimensionality, given we describe the space with a single categorical distribution. Error with respect to the closed form is shown on the y axis, for samples ranging from 100 to 10000 to give an idea of sample efficiency. Each line is the mean of 1000 runs of the simulation each with a different randomly generated normal. Note that our approach suffers from substantially less underestimation than traditional discretisation.

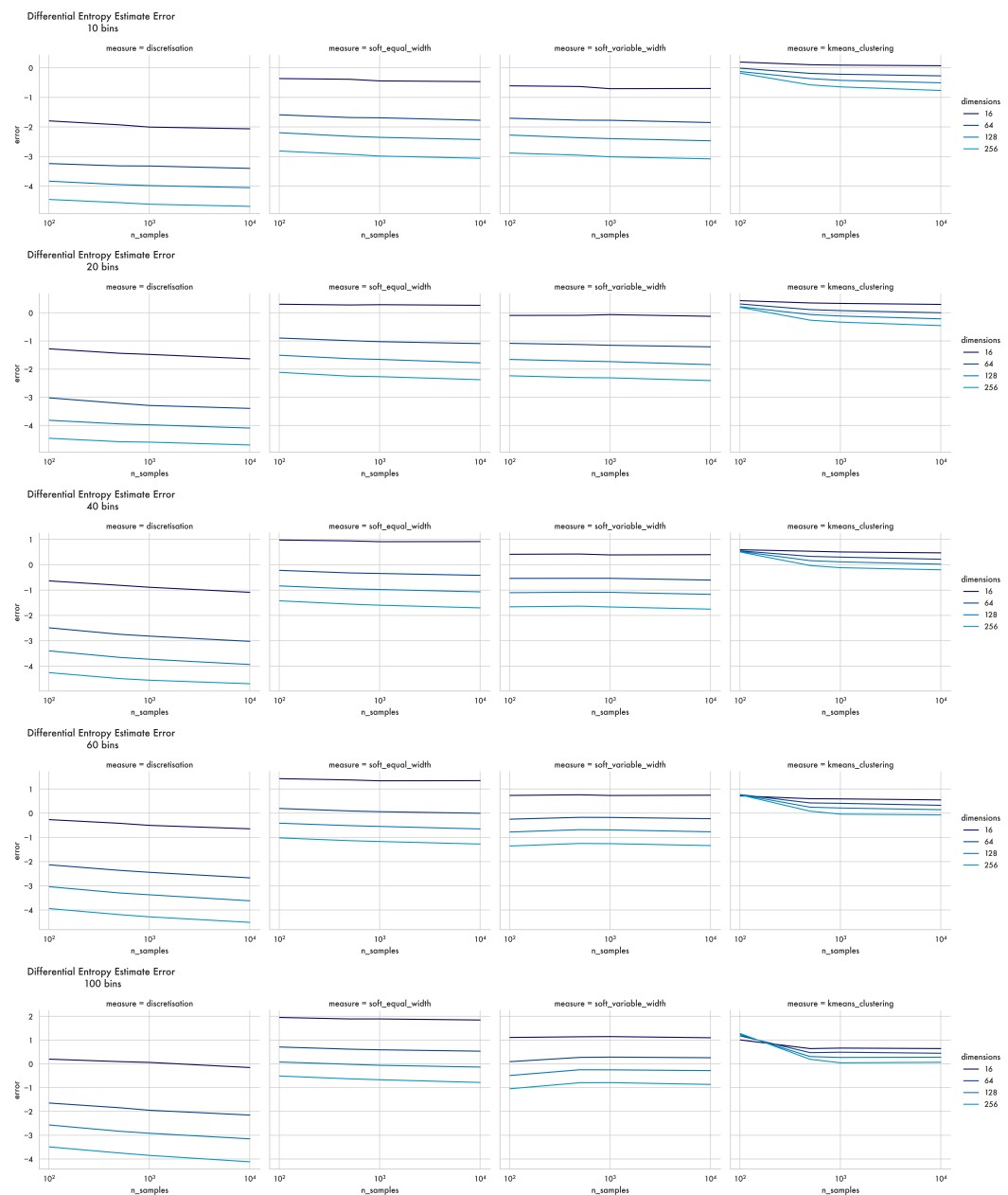

## A.2    BENCHMARKING: EFFECT OF NUMBER OF HEADS, MEAN AND SCALE

We compare versions of our estimator across different levels of subspacing. Angular entropy is the version that appears in the paper, discrete follows the methods of soft entropy estimation but then argmaxes to assign representations to a single point on the sphere. We also include a version that uses euclidean distances instead of the cosine-sphere comparison used in the paper. In principle this is nice because models also represent information topographically, encoding meaning in magnitude as well as angle in representational space. In practice euclidean distances end up being dramatically less memory efficient (and a factor of 4 slower to compute) than cosine similarities when using built-in pytorch methods. This means for scalability reasons we elected to only focus on the cosine case.

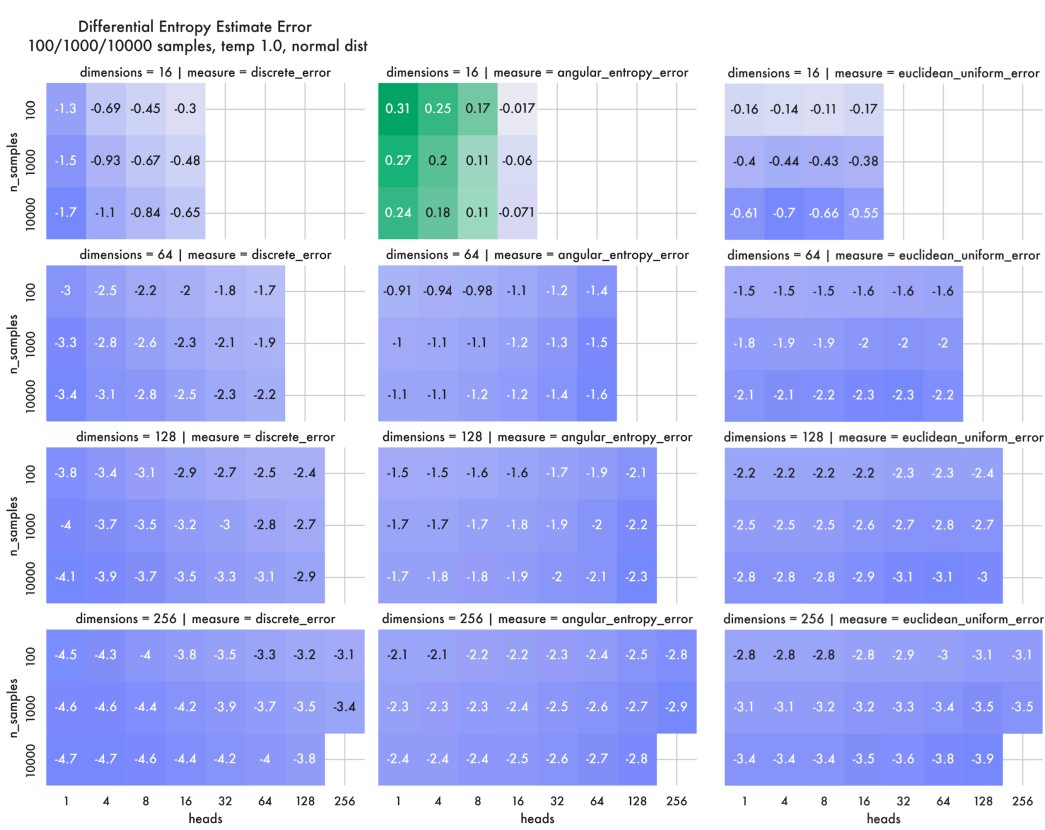

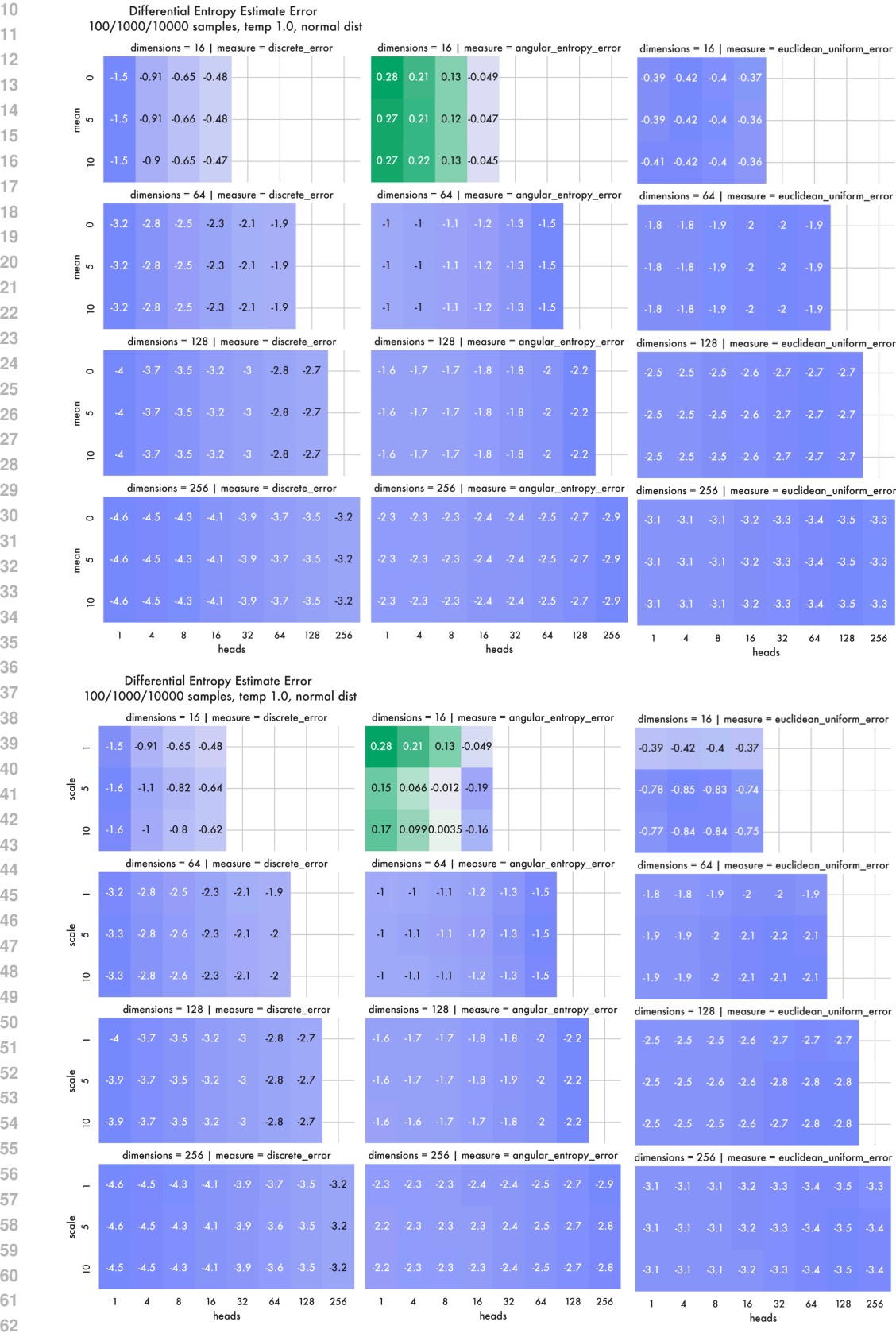

864
865
866
867
868
869
870
871
872
873
874
875
876
877
878
879
880
881
882
883
884
885
886
887
888
889
890
891
892
893
894
895
896
897
898
899
900
901
902
903
904
905
906
907
908
909
910
911
912
913
914
915
916
917

## A.3 ALL GLUE CORRELATIONS BY TASK

There are a huge volume of correlations for which I apologise

### QNLI Accuracy vs. Representational Structure

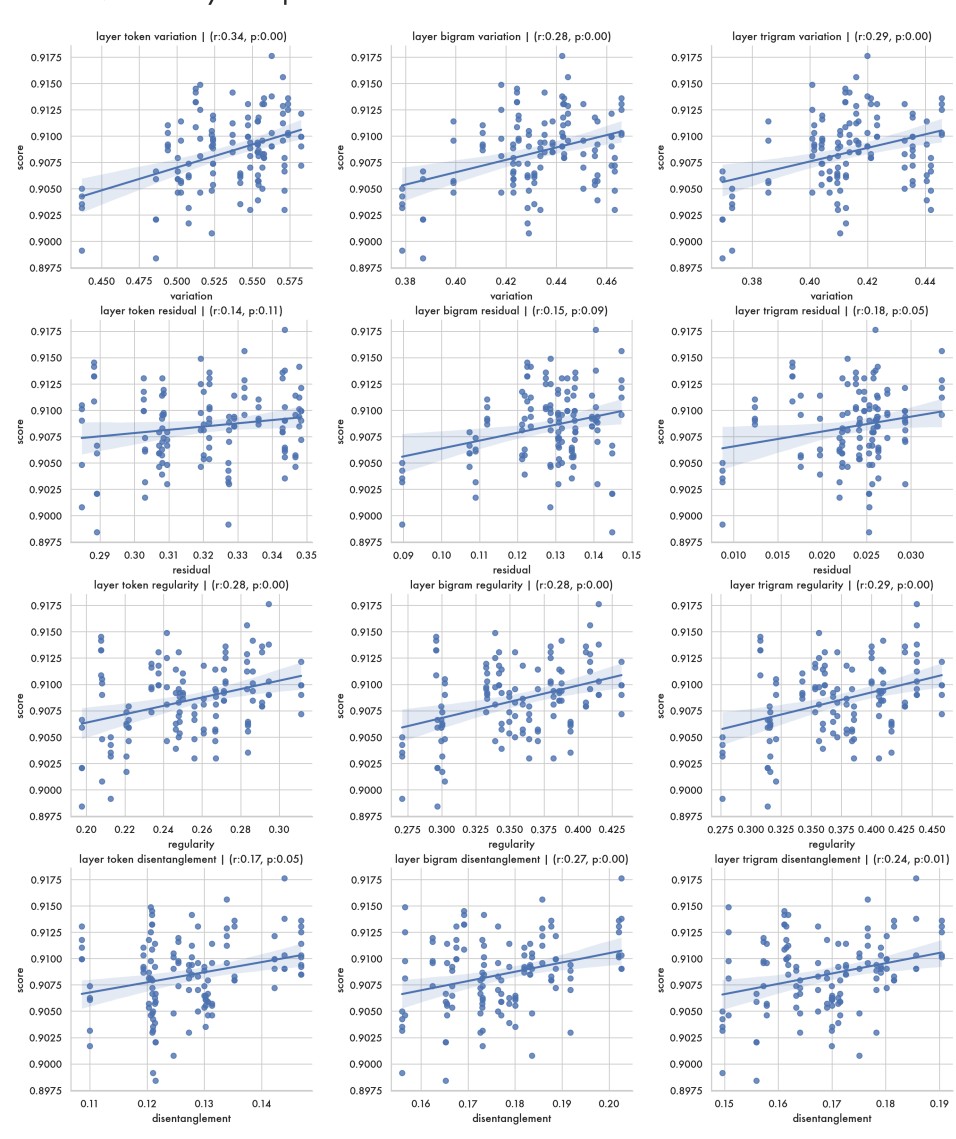

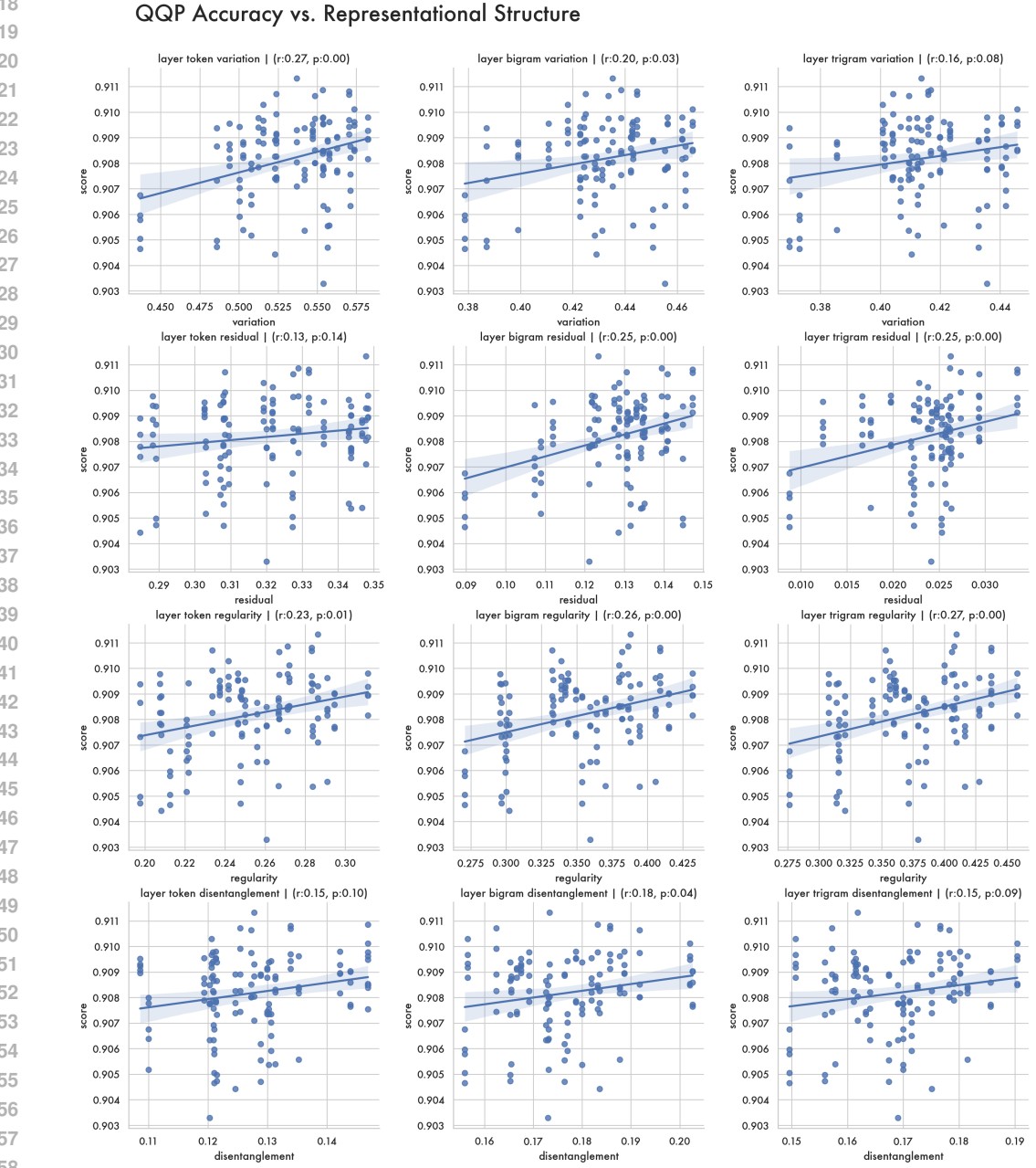

STS-B Accuracy vs. Representational Structure

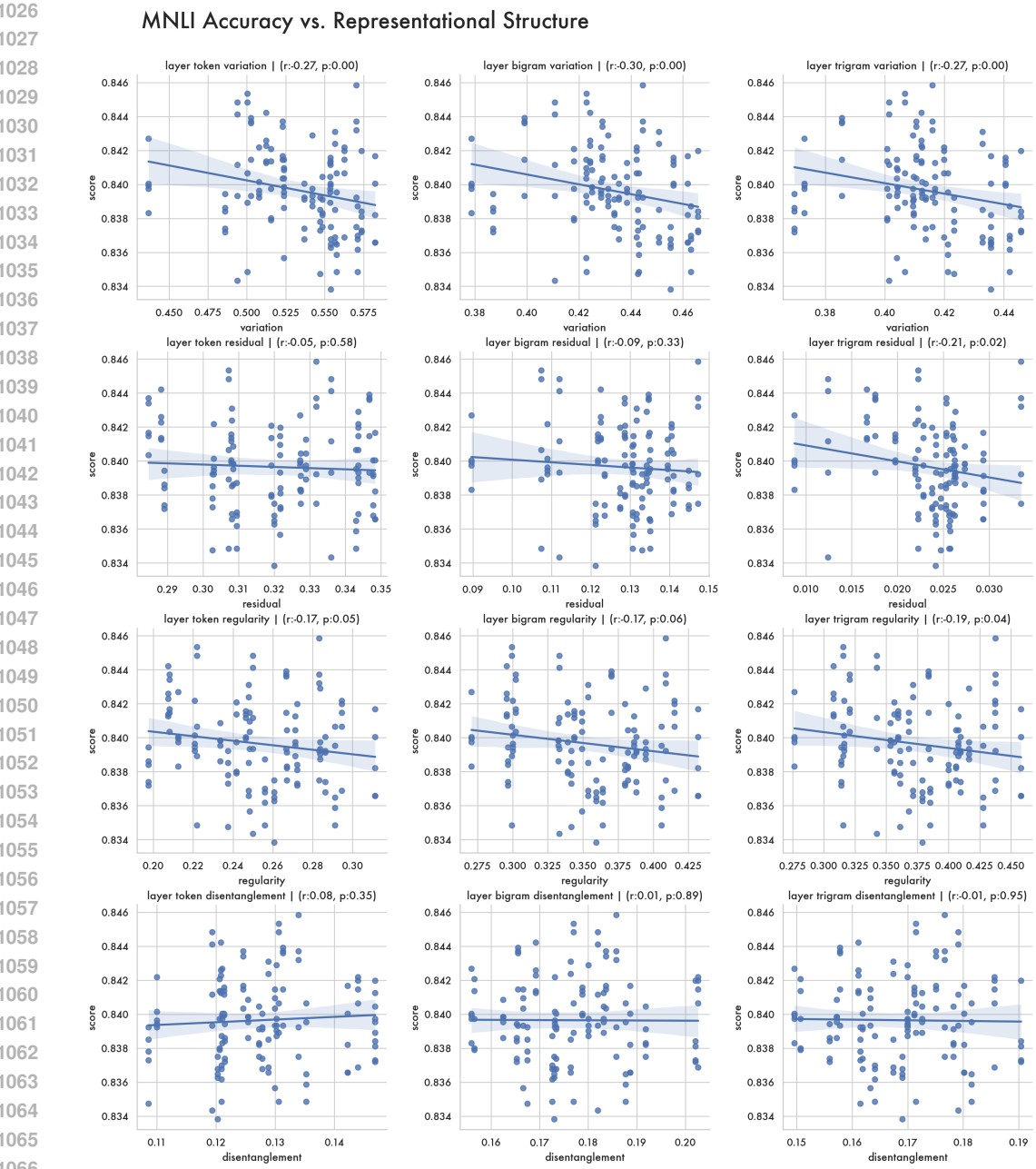

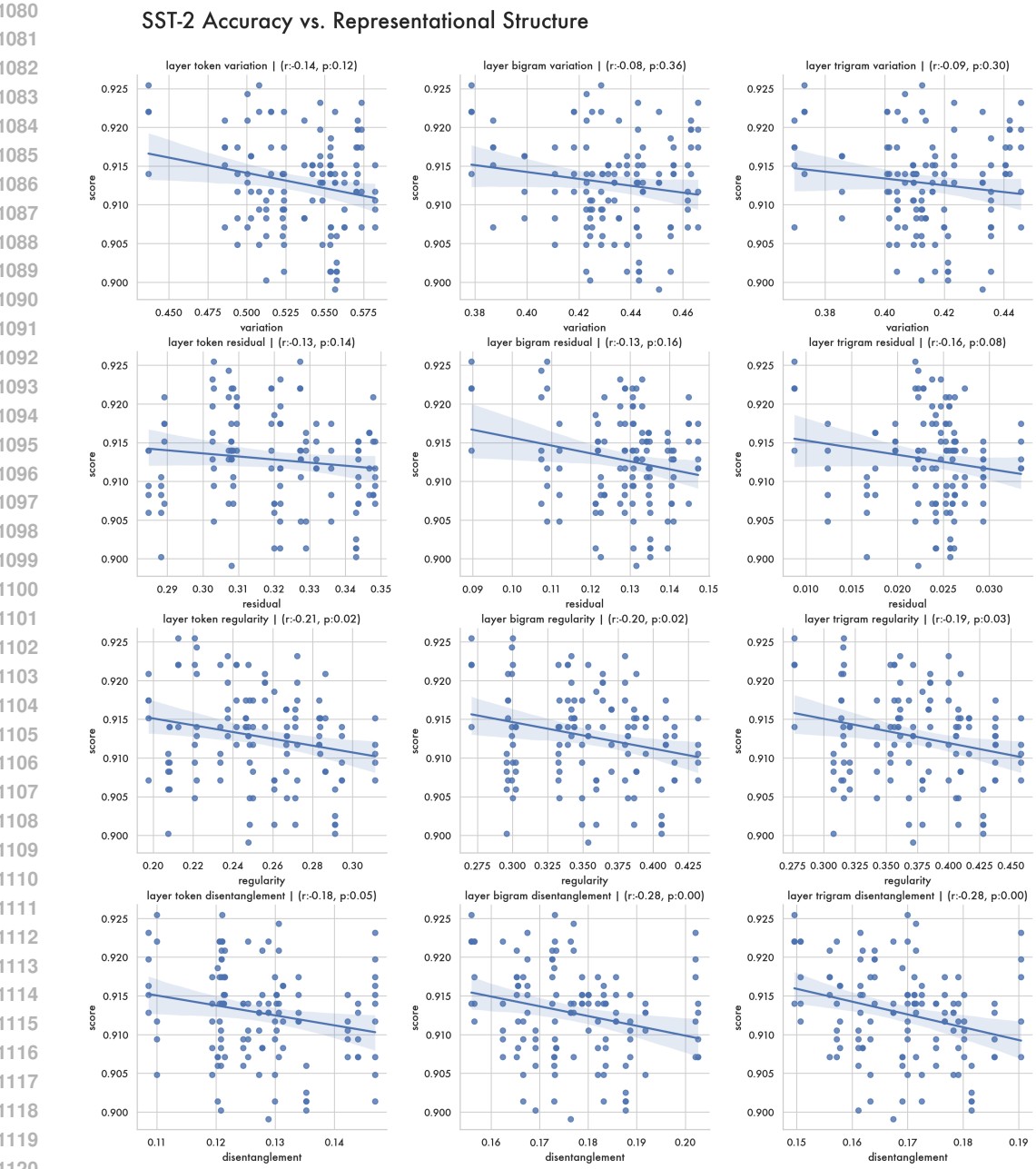

SST-2 Accuracy vs. Representational Structure

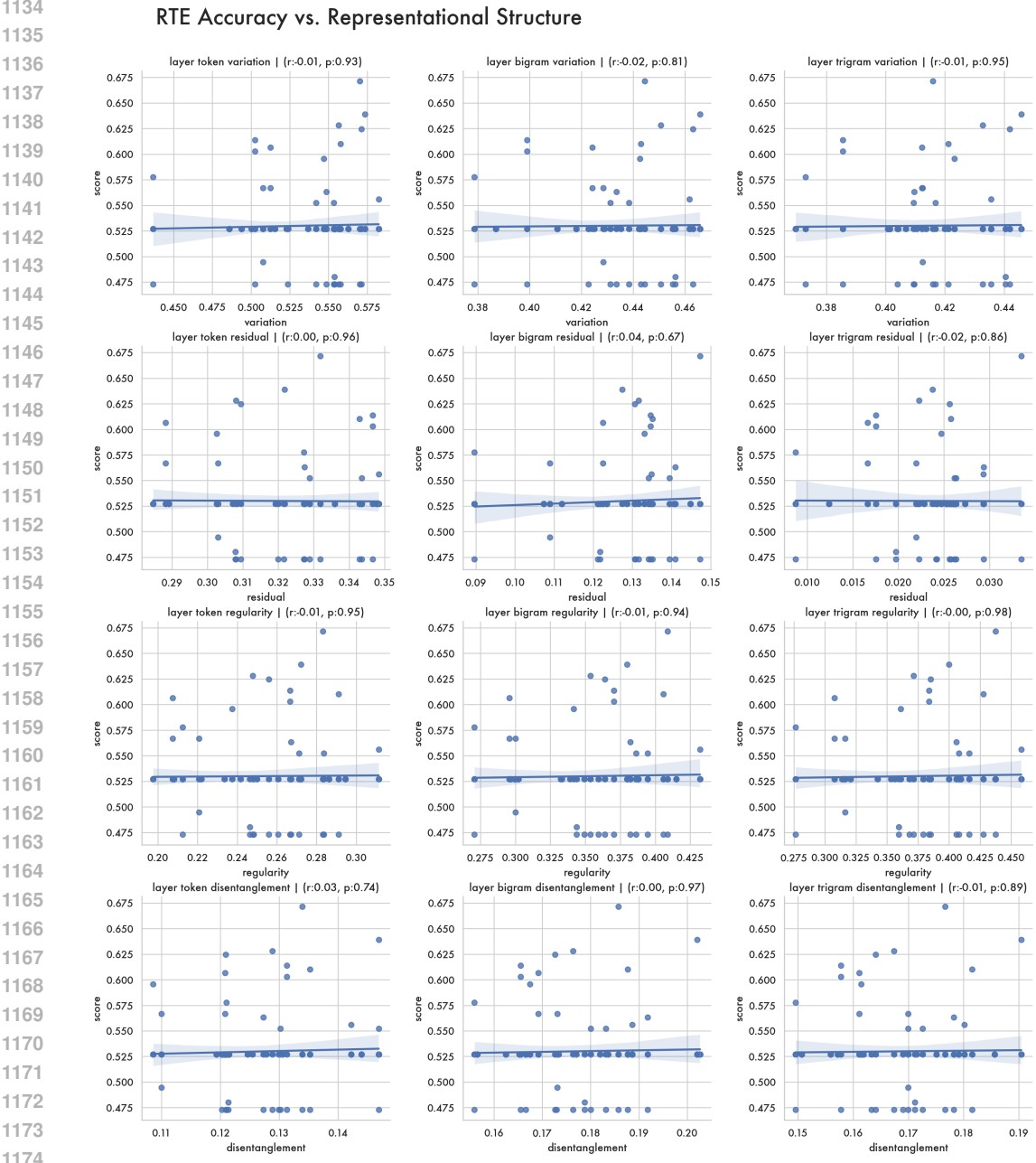

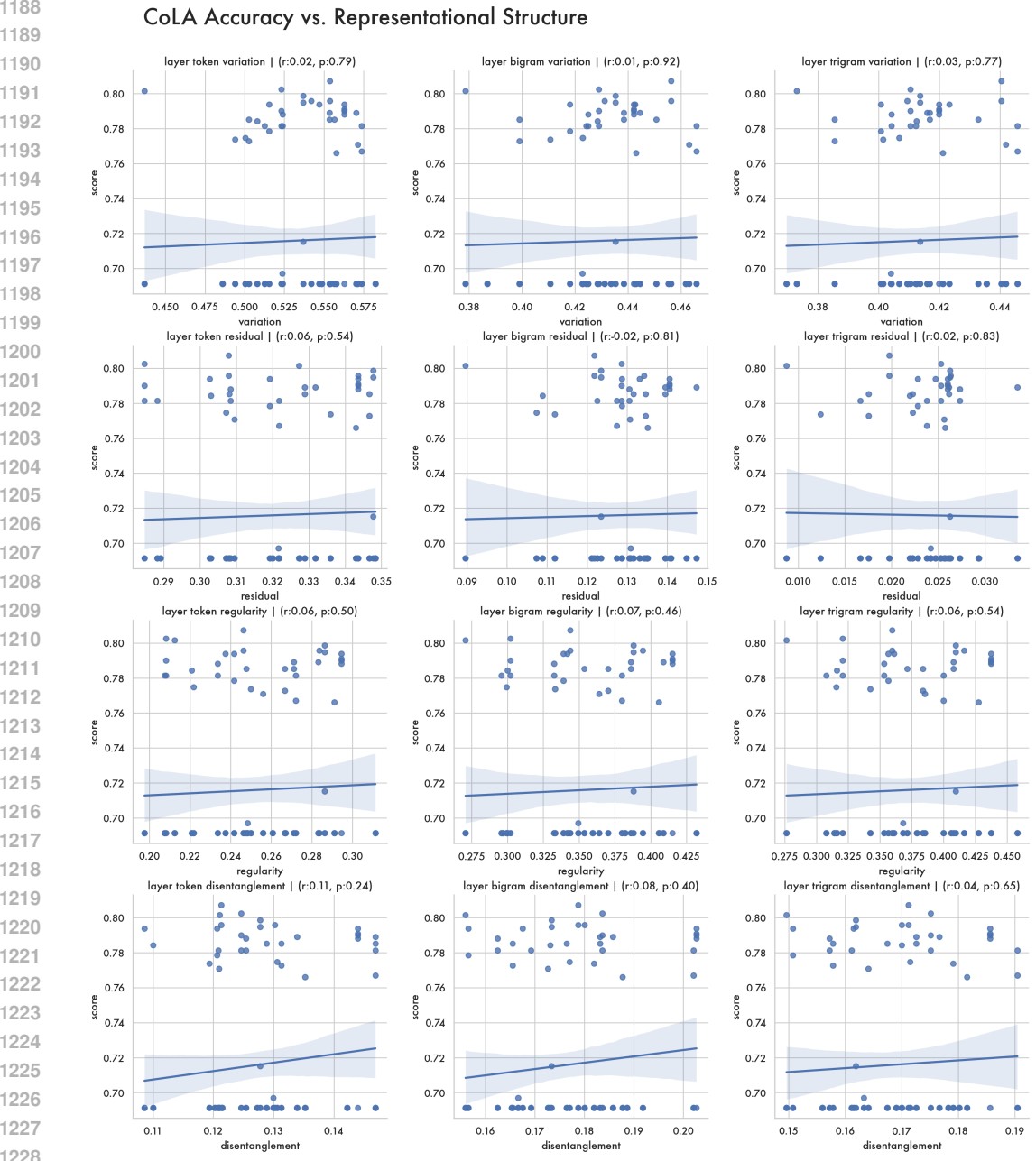

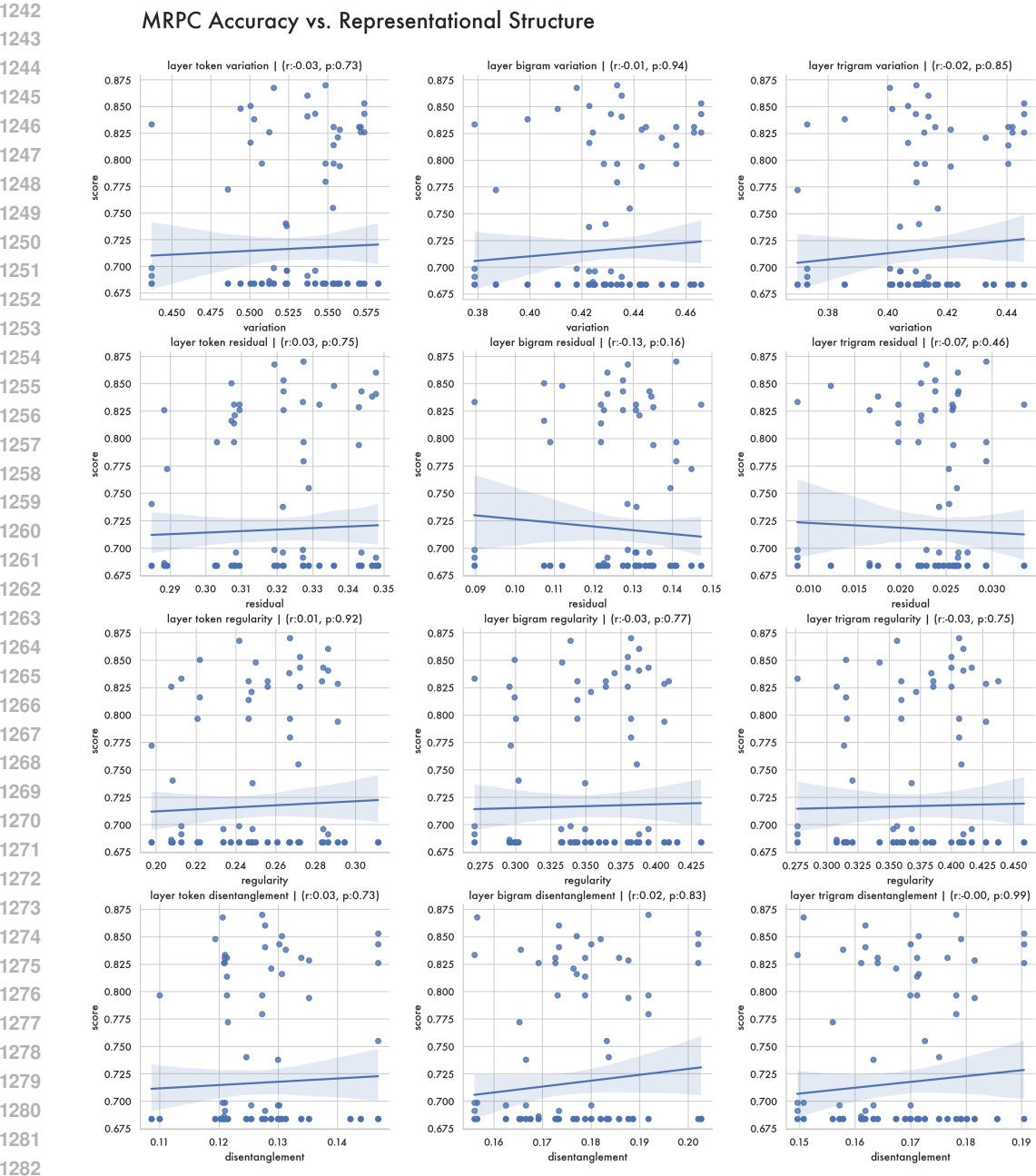

