# OpenReview forum: "Information Structure in Large Language Models"
_ICLR.cc/2025/Conference — Submitted to ICLR 2025_

### Official Review · Reviewer_yyPV · 2024-10-25

**Soundness:** 2
**Presentation:** 2
**Contribution:** 2
**Rating:** 3
**Confidence:** 4

**Summary:**

This paper, first, develops methods to quantify the structural properties of neural language models representations and, second, uses these methods to quantify said properties across a range of models.

For the first part, the paper focuses on three properties that they term "regularity", "disentanglement", and "variation". The authors then propose mathematical definitions for these quantities, which crucially rely on entropy estimates. To quantify entropy of a cloud of points in high dimensional space, which correspond to the neural network representations, they propose an estimator which they coin "soft entropy". This estimator is defined (I think because Eq. 5 is slightly underspecified) as the entropy of the average softmax-weighted cosine similarity of the representations to a sampled cloud of uniformly distributed points (where the average is across the representations).

For the second part, they look at these metrics over training time for a set of 5 BERT models, and over model sizes for the Pythia model suite. They also the computed features to try to predict downstream task performance on a set of 25 BERT models that only differ in their random initialization.

**Strengths:**

* S1: The paper's focus on characerizing the representations of large language models is of high interest to the community
* S2: The entropy estimator is very simple
* S3: The aim of predicting downstream task performance from the characteristics of the representations alone, even though I am not completely sure it is novel -- it certainly rings a bell, -- is quite neat.

**Weaknesses:**

* W1.a) The focus on the dimensions of "variation", "regularity" and "disentanglement" is not motivated enough. It is written that these are linguistic concepts, but no discussion is provided of what is their relative importance, and what do we learn from being able to characterize them.
* W1.b) The introduction motivates the conditional entropies in terms of denotational semantics, as in mappings between symbols and meanings, but then the analysis focuses on the mapping between vector representations and n-grams (n <= 3) , which is a mapping between to representational levels.
* W2: The definitions of these concepts around different entropy measures is also quite arbitrary with little justification of why it should take this particular form.
* W3: Similarly, the entropy estimator comes with no theoretical analysis nor motivation.
* W4: At the end of the day, the analyses focuses on the variability around n-gram representations, which is (and I acknowledge this is subjective) not very illuminating of the internal structure of the models.
* W5: The correlation of representational structure to the down-stream performance of the fine-tuned models, however neat is the aim, seems to yield nothing. While it is true that there is at least one task where it has shown some positive correlation, there are many others where it didn't or where it showed negative correlations. It is loable that the authors did not hide this under the rug, but cherry-picking the one task where it did work is not sound either. At the very least, some correction for multiple comparisons should have been made to these correlations when testing for statistical significance.

All in all, W1, W2/W3, W4, and W5, respectively, undermine each of the claimed contributions of this paper.

* W6: Furthermore, the paper was not very easy to read, as it often lacks structure. I cannot give very precise examples without becoming too extensive, but, for example, the introduction is vague and it takes a long time before one starts to learn what this paper is really about. Also, the mathematical notation is a bit messed up. Take, for example, the summations in equations 2, 3 and 4 which have a very non-standard form. Eq. 5 misses the subindex of the summation. Etc.

**Questions:**

* Q1) Could you clarify what you mean by system level structures and in which way your study helps us elucidate them?
* Q2) More importantly, what have we learned from this study that we didn't know before about these models?
* Q3) Is it not the case that your estimator is very sensitive to the vector dimensionality? I understand that as I increase the dimensionality the chances of vectors just being orthogonal to each other increases by a large margin. Wouldn't I need to sample way more points as I compute the entropy in larger dimensions? Could you test this by, say, generating samples with known fixed entropy, increase the dimensionality of these samples and see how the error scales?
* Q4) Since Elman (1990), SVD/PCA has been one of the weapons of choice to analyze representational structure of these models. What does your framework capture that spectral methods don't?
* Q5) When you talk about heads in Sec. 4.1, which you then use to define subspace entropy, do I get right that these "heads" are chops of the vector at arbitrary position and have nothing to do with attention heads or are they actually aligned with them? If the latter, how do you deal with models with different head dimensionality?

Here are some typos, clarifications, and other suggestions
* L034 effect -> affect representational space
* L172: How do you bound the metric to the [0,1] range?
* L195: you claim it is faster, and I believe you, but numbers would help drive the point
* L240: mean across -> average across
* L262: rescale the distances -> which distances? wheren't these cosines in the [-1, 1] range?
* L299: aggregated how?
* In Figure 3 middle, is there a typo in the first row where it should be "layer" rather than "subspace"?

---

### Official Review · Reviewer_UT2F · 2024-11-02

**Soundness:** 2
**Presentation:** 1
**Contribution:** 2
**Rating:** 3
**Confidence:** 3

**Summary:**

This paper proposes to use measures based on information theory to assess high-dimensional representations learned by LLMs. These measures --- variation, regularity, disentanglement --- relate linguistic features of the texts --- in the case of this paper, unigram, bigram and trigram features associated to each token in the text --- to the vector representations learned by the model.

The main novel contributions of the paper relative to related work cited, in particular [CS2024], appears to be (1) the proposal of a ``soft entropy'' estimation procedure, argued to provide efficiency and differentiability advantages, and (2) extended experiments over a variety of BERT models, including assessment of correlations between the information-theoretic measures and downstream performance over GLUE benchmarks.


[CS2024] Conklin and Smith, 2024. Representations as language: An information-theoretic framework for interpretability.

**Strengths:**

The paper advocates for the use of interesting information-theoretic measures that allow to assess the dynamics of LLM learning by computing various entropy metrics that relate linguistic structure to high-dimensional representations (such measures were introduced in [CS2024]).

It proposes estimations of these measures based on a technique that does not require the binning of each dimension of the vector representation, but is based on the calculation of a cosine similarity between the full vector and $n$ reference points placed on the unit sphere in vector space. This is a "soft'', and hence differentiable measure, contrary to bin-based measures that can be non-differentiable at the boundary between bins (my understanding).

It also proposes a number of experiments based on BERT-trained models, with variants of the measures at the "layer" or "subspace" levels, and assesses correlations between the information-theoretic measures and downstream tasks over GLUE.

**Weaknesses:**

The main weaknesses of the paper are at the following levels: (1) differentiation with related work, (2) clarity and self-containment of the description of the information measures, (3)  formal quality of the explanations about soft entropy estimation.

(1) Differentiation relative to [CS2024]. The authors do cite this arXiv paper in their related work section, but do not clarify the very large amount of overlap with it, in particular the fact that all the information measures they propose were already described in it, and IMO in a much clearer way than what is done here. Related to this point, the title of the submission "Information Structure in Large Language Models" could be reformulated to better reflect the actual differentiation with [CS2024].

(2) Clarity and self-containment of the descriptions. This is related to the previous item. At different points during my reading of this submission, for instance when looking at Fig. 1, or when reading the formal descriptions of the different information measures, I found the description mathematically confusing and had to refer to [CS2024] for a better understanding. To give only two examples: (i) $Z$ is defined as a set, but then the authors speak of the entropy $H(Z)$ of this set, while the entropy is a property of probability distributions, (ii) the explanation of disentanglement in terms of Jensen-Shannon divergence, both in Fig.1 and in the text, was unclear to me, and would need to be related to the usual notion (the Appendix could be used for that, as for other formal details lacking in the paper). Unfortunately, the problem with the mathematical notations and explanations extends over much of section 3.

(3)  Soft-entropy estimation is an important differentiation point with respect to [CS2024]. While the underlying intuition is interesting and relatively clear (see Strengths above), the mathematical formalization in section 4.1 is quite unclear (in particular I was not able to parse equation (5)). See also the related point in the Questions below.

**Questions:**

Question 1.1, about soft entropy estimation. If I try to reconstruct what you do, once you have produced the $n$ ``anchor points'' on the unit sphere --- BTW, it is not clear exactly how you do that, and you could cite some reference about how to do that, for instance [Muller1959] --- you take each point $z$  in $Z$, then compute the cosine similarity between $z$ and each of the $n$ reference points, apply a softmax over this vector, obtaining a new vector of dimension $n$ whose coordinates sum to 1. You then get a set of $|Z|$ such vectors, of which you take the mean, which can be seen as a distribution over the $n$ points. You then compute the entropy of this distribution.
Is that correct ?

Question 1.2. Assuming what I said above is correct, suppose that all the $z$'s in $Z$ are actually the same point. Then you would expect the entropy associated with $Z$ to be 0, or very close to 0. But I think the soft-entropy as you compute it has no reason to be close to 0. Do you agree? And is that a problem for your approach to soft entropy ?

Question 2. At the bottom of Figure 3, one of your measures is "efficiency (normalised entropy)", which you do not explicitly define (unless I missed it), but which I understand is a way to obtain a measure that is more comparable among models of different complexities. Would such a measure be adequate for a fair comparison of the "layer entropies" across the different models you consider in the figure, without considering the more focussed "subspace entropies"?

[Muller1959] A note on a method for generating points uniformly on n-dimensional spheres. CACM 1959.

---

### Official Review · Reviewer_6NF6 · 2024-11-04

**Soundness:** 3
**Presentation:** 2
**Contribution:** 2
**Rating:** 5
**Confidence:** 3

**Summary:**

The paper explores an information-theoretic framework to understand the structure of representation spaces in language models. The authors propose a method for estimating entropy, called "soft entropy". This method quantifies the degree of regularity, variation, and disentanglement in a high dimensional representation space, building off of the structure of mappings (one-to-one, many-to-one) to do their analysis.

Key findings include that larger models tend to exhibit greater disentanglement and contextual encoding capacity, which correlates with improved downstream task performance (demonstrated on the GLUE benchmark). The study also highlights how representational structures emerge and evolve during training, noting that contextual representations become more prominent as training progresses. This framework offers a quantitative approach to predicting model performance based on pre-training representational structure and aims to enhance the interpretability and design of LLMs.

**Strengths:**

Having simple methods for model interpretability is very important for the field, especially when those methods don't rely on specific datasets or manual analysis. The work here would work with any generic text corpus. This work reminds me of the weight watchers tool, which evaluates how well trained model weights are without running them (https://github.com/CalculatedContent/WeightWatcher). Whereas this paper uses entropy calculations over model activations

Predicting downstream accuracy is a very interesting application of the methods presented here

**Weaknesses:**

Not clear what we learn from the training time experiments. The initial change and then smooth degradation does not seem to follow a more interesting pattern

short related work

To make claims about predicting benchmark performance, usually a regression would be taken but only correlation results are included here

There is an issue with the writing. Terms are introduced without definition (L85 system level), for example. L266 runon sentence.

**Questions:**

N/A

---

### Official Review · Reviewer_AuzR · 2024-11-04

**Soundness:** 2
**Presentation:** 1
**Contribution:** 2
**Rating:** 3
**Confidence:** 4

**Summary:**

This paper proposes an information-theoretic approach to measure the structures of learned function mappings in neural language models, quantified as entropy of the distribution of hidden representations in language model activation space. The paper estimates embedding space entropy as the divergence between the empirical normalized embedding distribution and a uniform distribution on a unit sphere. The paper then shows that their proposed entropy estimator can be applied to reveal emerging embedding space structures during pretraining that correspond to linguistic properties of regularity, disentanglement, and variation. They also show that their estimated structural properties of representation space are correlated to model performance on some downstream NLU tasks.

**Strengths:**

The paper proposes using an information-theoretic approach to study the evolution of learned function mapping in neural language models. It confirms that language models gradually learn structured representation space during pretraining with low intrinsic dimension, which was generally agreed.

**Weaknesses:**

1. Some of the proposed methods and findings are not novel. For instance, the entropy estimation method resembles those proposed by (Conklin & Smith, 2024) and (Saxe et a;., 2019). The definitions of regularity, disentanglement, and variation also duplicate the notions introduced by (Conklin & Smith, 2024). The demonstrated two-phase structural transition of LM embedding space also mirrors previous findings by (Conklin & Smith, 2024), (Shwartz-Ziv & Tishby, 2017).

2. The formulation of variation, regularity, disentanglement, and soft entropy estimations are sketchy, missing sufficient justifications of the design choices.

3. The evaluation experiments are not complete. In particular, the demonstrated two-phase transition and compressed dimensionality could be due to many factors like model architecture, training data, and training loss functions. The authors should conduct a more systematic ablation study to show the effect of these factors  -- for example, what if we train BERT using "corrupted" text data (e.g., with word order randomly shuffled or replaced by nonsensical tokens)?

4. The writing in this article lacks rigor; many of the expressions in the main text are colloquial or imprecise.

**Questions:**

The notions of variation, regularity and disentanglement still seem a bit vague and inaccessible to me, could the authors clarify these concepts through a few illustrative examples in future revisions?

---

### Meta-Review · Area_Chair_gpcE · 2024-12-20

**Metareview:**

This paper explores an information-theoretic view on the structure of representation spaces in language models. Reviewers had strong reservations about both the clarity of the paper and its relationship with other work. There was a clear consensus that this paper is not yet ready for publication.

**Additional Comments On Reviewer Discussion:**

See above

---

### Decision · Program_Chairs · 2025-01-22

Reject